# Calprotectin, Biomarker of Depression in Patients with Inflammatory Bowel Disease?

**DOI:** 10.3390/medicina59071240

**Published:** 2023-07-03

**Authors:** Miorita Melina Iordache, Anca Mihaela Belu, Sabina E. Vlad, Kamer Ainur Aivaz, Andrei Dumitru, Cristina Tocia, Eugen Dumitru

**Affiliations:** 1Faculty of Medicine, Ovidius University of Constanta, 1 Universitatii Alley, 900470 Constanta, Romania; meliordache@gmail.com (M.M.I.); dr.andreidumitru@gmail.com (A.D.); cristina.tocia@yahoo.com (C.T.); eugen.dumitru@yahoo.com (E.D.); 2Prof. Alexandru Obregia Psychiatry Hospital, 10 Berceni Str., 041914 Bucharest, Romania; 3“St. Apostol Andrew” Emergency County Hospital, 145 Tomis Blvd., 900591 Constanta, Romania; 4Center for Research and Development of the Morphological and Genetic Studies of Malignant Pathology-CEDMOG, “Ovidius” University of Constanta, 900591 Constanta, Romania; sabinaochiana@gmail.com; 5Faculty of Economics, Ovidius University of Constanta, 1 Universitatii Street, 900470 Constanta, Romania; aivaz_kamer@yahoo.com; 6Academy of Romanian Scientists, 3 Ilfov Street, 050045 Bucharest, Romania

**Keywords:** mental disorders, biomarkers, S100A8/A9 proteins, suicidal ideation, fatigue, sleep disorders

## Abstract

*Background and Objectives*: Calprotectin is a marker for intestinal inflammation. Recent research suggests a link between inflammation and depression. This study assessed the association between the levels of calprotectin in patients from South-Eastern Europe and the severity of depression, anxiety, and quality of life. *Materials and Methods*: This cross-sectional study included 30 confirmed patients with Crohn’s disease (CD) and ulcerative colitis (UC) who were assessed using clinical interviews for determining the severities of mental disorders (i.e., depression severity—PHQ-9, anxiety—GAD-7) and the quality of life (EQ-5D). Stool samples were collected from all participants for measuring their levels of calprotectin. *Results*: The level of calprotectin is correlated with PHQ-9 (ρ = 0.416, *p* = 0.022) and EQ-5D (ρ = −0.304, *p* = 0.033) but not with GAD 7 *(*ρ = 0.059, *p* = 0.379). Calprotectin levels in patients with mild, moderate, and moderately severe depression were significantly higher than in patients with minimal depression (198 µg/g vs. 66,9 µg/g, *p* = 0.04). Calprotectin level was corelated with the following depressive symptoms: autolytic ideation (ρ = 0.557, *p* = 0.001), fatigue (ρ = 0.514, *p* = 0.002), slow movement (ρ = 0.490, *p* = 0.003), and sleep disorders (ρ = 0.403, *p* = 0.014). Calprotectin was an independent predictor of depression with an odds ratio of 1.01 (95%: 1.002–1.03, *p* < 0.01). An ROC analysis showed that a level of calprotectin of 131 µg/g or higher has a sensitivity of 82%, a specificity of 61%, and an accuracy of 70% for predicting depression. In this study, no significant correlations were found between calprotectin level and anxiety. *Conclusions*: Calprotectin levels are associated with the severity of depression, and checking for a calprotectin level of 131 µg/g or higher may be a potential accessible screening test for depression in patients with inflammatory bowel disease.

## 1. Introduction

Inflammatory bowel disease (IBD) includes Crohn’s disease (CD) and ulcerative colitis (UC), both of which are characterized by the chronic inflammation of the gastrointestinal tract. Hence, these are associated with gastrointestinal and extra-intestinal clinical manifestations. In 2019, there were 4.9 million IBD cases worldwide [1]. The combined prevalence of the two conditions in the population from Western Europe is of 450/10^5^, in the US 245.3/10^5^, in China 66.9/10^5^, in Japan 165.1/10^5^, and in the Republic of Korea 108.4/10^5^. Their incidence is increasing in the emerging countries of Africa, Asia, and South America [2,3].

Calprotectin is a calcium-binding protein of the S100 family and an inflammatory marker with regulatory functions in inflammatory processes and antimicrobial and antiproliferative activity [4]. Thus, calprotectin is a marker for significant neutrophilic bowel inflammation [5]. The diagnosis of IBD, along with clinical, endoscopic, imaging, and histological criteria, uses calprotectin as a non-invasive test [4].

In a meta-analysis, Van Rheenen stated that elevated stool calprotectin levels contribute to identifying the patients most likely to have IBD [6]. Furthermore, calprotectin could discriminate endoscopic inactivity by mild, moderate, and severe disease activity, in a manner superior to clinical indices such as the erythrocyte sedimentation rate or leukocytes [7,8]. It also has relative sensitivity and specificity rates in estimating recurrence and response treatment [9]. Its diagnostic performance is more reliable for UC than CD [10], but it does not distinguish between the two conditions and does not allow disease localization [11].

Yet, elevated calprotectin levels in stool cannot be exclusively attributed to IBD, since such increases can be also associated with other diseases—for e.g., inflammatory (ankylosing spondylitis, systemic sclerosis, diverticular disease, celiac disease), infectious (AIDS, infectious diarrhea), neoplastic (colon cancer, pancreatic cancer), and iatrogenic (nonsteroidal anti-inflammatory drugs and proton pump inhibitors, radiotherapy) diseases, as with various other factors (food allergy, young age) [12].

Since the gut–brain axis has a two-way communication, it also provides resources for the study of mental health [13]. Patients with IBD have psychiatric comorbidities that significantly impair their quality of life [14].

Depression is characterized by two fundamental elements: lack of pleasure/interest in previously pleasurable activities and feelings of sadness, with symptoms lasting longer than two weeks [15]. The average prevalence of depression in Europe was 7%, with variations between regions [16]—for e.g., 4.75% in Western Europe [17] and 2.13% in Central andEastern Europe [18]. There are also large differences worldwide: 9.2% in the USA [19] and 26.4% in Asia [20]. The pooled mean rate of depressive symptoms in CT IBD is 21.2% with a higher prevalence in CD (25.3%) than in UC (16.7%) and a higher prevalence in active (40.7%) than inactive disease (6.5%) [21]. Depressed patients had more aggressive illnesses at the time of diagnosis, and earlier, than patients who were not depressed, and a scale was used to identify those at higher risk of illness over time [22].

Depression is frequently associated with anxiety, and both influence a low quality of life. The symptoms of depression and anxiety have far-reaching implications for IBD. Their treatment should be included in managing IBD [23].

The key is therapeutic adherence, which is influenced by depressive symptoms and impacting disease remission, low recurrence rates, the prevention of relapses, and higher demand of medical services [24]. Since these can have serious consequences, it is imperative to better understand how to quickly diagnose depression in IBD. Having a greater insight into depression screening would constitute useful information for early intervention in depression.

The depression diagnostic is clinical, and it misses the usual biological biomarkers. The lack of biomarkers in depression constitutes a new domain with unstudied potential. Understanding the correlations between depression and intestinal biomarkers is important not only for patients with depression and IBD but also for patients with depression in general.

The current study continues our previous research, investigating the relationship between psychiatric disorders and the gut–brain axis, mainly related to intestinal permeability syndrome [25]. Therefore, in this paper we aim to assess (i) the correlations between calprotectin level and depression, anxiety, and quality of life, based on patients’ responses to interviews. Further, we evaluated the (ii) correlations between calprotectin levels and depression severity. We investigated whether (iii) depression symptoms are correlated with calprotectin. Finally, we (iv) provided a cut-off value for calprotectin which can be informative in the screening for depression.

## 2. Materials and Methods

### 2.1. Target Group

The study was conducted in the Ambulatory of the Constanta County Hospital between 1 April and 30 June 2021. The hospital serves a local population of 750,000 inhabitants. Patients over 18 years of age with radiological, histological, or endoscopic diagnosis of CD or UC undergoing treatment in secondary care were invited to the study. A psychiatrist conducted a structured interview and screened for depression severity, anxiety, and quality of life. The exclusion criteria were: severe active IBD, severe psychiatric comorbidities (schizophrenia, dementia), Patient Health Questionnaire-9 scores over 19, and the lack of informed consent. Out of 60 patients, 30 patients who gave informed consent and were in accordance with the inclusion criteria were included in the study.

### 2.2. Biological Samples

The protocol for collecting and transporting potentially infectious biological samples was respected regarding fecal calprotectin samples. The samples were collected from spontaneously passed stool in containers without carrier medium and stored at −20 °C until the day of testing. Stool sample extraction was performed using 980 UI extraction buffer and 20 mg stool, stirred for 30 s, and then centrifuged for 10 min at 2000× *g* using the Thermo Scientific SL16R centrifuge (Thermo Fisher Scientific Inc., Osterode am Harz, Germany).

The tests were performed using the Enzyme-Related Immunosorbent Test (ELISA) on an ADALTIS GEN-4 and Victor X4 analyzer according to the instructions of the manufacturer of the kit from EUROIMMUN Medizinische Labor-Diagnostika. A number of 96-well microplates coated with capture antibodies were used; these antibodies were anti-calprotectin antibodies. The range of normal values for calprotectin was defined by the manufacturer of the kit as follows: ≤50.0 µg/g as usual, 50.1–120.0 µg/g as a limit value, and ≥120.1 µg/g as positive for IBD.

### 2.3. Applied Questionnaires

For this study, three validated questionnaires were used, i.e., The Patient Health Questionnaire-9 (PHQ-9), General Anxiety Disorders questionnaire (GAD-7), and Quality of Life questionnaire (EQ-5D).

*The Patient Health Questionnaire-9* is an accessible and concise tool which aims to diagnose and assess the severity of depressive disorders. It has nine items correlated with the diagnostic criteria for depressive disorders provided by DSM-IV. There are cognitive, emotional, and somatic symptoms of depressive disorders. For diagnosis, five criteria or more are required. It is mandatory that one of the criteria is depressed mood or anhedonia. Depressive symptoms should be present for more than two weeks, “more than half the days.” The criterion of autolytic ideation (“thoughts that it would be better to die or hurt yourself in some way”) matters if it is present, regardless of duration [26].

For minimal depression, the score is between 1–4, mild depression 5–9, moderate depression 10–14, moderately severe depression between 15–19, and severe depression 20–27. Construct validity was assessed using the 20 items of the Short-Form General Health Survey (SF-20), self-reported sick days, and clinic visits. Criterion validities were assessed with an independent and structured mental health professional (MHP) interview [26].

The internal reliability of the PHQ-9 was excellent with a Cronbach’s α of 0.89 [26]. Test–retest reliability of the PHQ-9 indicated consistency over time. The correlation between patient self-reported PHQ-9 and the score of PHQ-9 administered telephonically by a MHP within 48 h was 0.084. Receiver operating characteristic (ROC) analysis demonstrated the high discriminative ability of the PHQ-9 in diagnosing major depression, with an area under the curve (AUC) of 0.95. For diagnosing major depression, the PHQ-9 showed a sensitivity of 84%, specificity of 72%, and a positive likelihood ratio of 2.86. These values indicate the test’s ability to accurately identify individuals with major depression. The studies included a substantial sample size of 6000 patients [26].

The *General Anxiety Disorders questionnaire* is a valid and efficient tool for screening and assessing the severity of anxiety in clinical practice and research and in evaluating the effectiveness of therapeutic interventions. It reflects the criteria of the Diagnostic and Statistical Manual of Mental Disorders, Fourth Edition (DSM-IV), which give it the highest degree of sensitivity, specificity, and validity. It consists of 7 questions, each with 4 options, that reflect the frequency and severity of anxiety symptoms: ‘not at all‘, ‘several days‘, ‘more than half the days‘, and ‘almost every day‘. The scores are: minimal anxiety between 0–4, low anxiety between 5–9, moderate anxiety between 10–14, and severe anxiety between 15–21 [27].

The criteria and construct validity were assessed using the Short-Form General Health Survey (SF-20) and the Beck Anxiety Inventory. Self-reported sick days, clinic visits, and diagnoses made by mental health professionals were also considered.

GAD-7 has good reliability, with internal consistency (Cronbach’s α = 0.92) and test–retest reliability (intraclass correlation = 0.83). A cut-point of 10 and more had a sensitivity (89%) and specificity (82%). The studies included a substantial sample size of 1184 participants [27].

*Health-related quality of life* reflects participants’ perceptions of their quality of life. It involves five items: patient mobility, self-care, participation in daily activities, pain or discomfort, and experienced anxiety or depression, with 5 answer options. The EQ-5D questionnaire also includes an analog visual scale (VAS) through which each respondent can report their perceived health status with a grade ranging from 0 (worst possible health) to 100 (best possible health) [28].

The PHQ-9 and GAD-7 scales have been used in studies that have looked at the prevalence of depression and anxiety in patients with IBD [29]. The Patient Health Questionnaire-9 was used as an algorithm for the potential diagnosis of major depression using a cut-off point of ≥10 [30]. PHQ-9 has also been shown to be a valid tool in screening for autolytic ideation in IBD patients [31].

### 2.4. Statistical Analysis

Data analysis was performed using the IBM SPSS software, version 23 [32]. The Shapiro–Wilk normality test indicated nonparametric variables. Accordingly, we did not provide mean values, but instead, median values and the interquartile range [33]. Descriptive statistics were generated for all continuous variables.

For measuring the degree of association between calprotectin, PHQ-9, GAD-7, and EQ-5D, we used Spearman’s correlation coefficient rho (ρ) and its associated probability (*p*). The Mann–Whitney test (U) assessed the difference between the groups.

Receiver operating characteristic (ROC) analysis was performed for evaluating calprotectin screening performance in R using Rbase version 4.2.3. Principal component analysis (PCA) was used to analyze the relationship between the variables (symptoms of depression and anxiety), observe the similarities and differences between statistical units, and allow for the identification of clusters. A value of *p* < 0.05 was considered statistically significant.

## 3. Results

### 3.1. Clinical and Demographic Characteristics

In the present study the gender ratio was balanced. The average age of the patients in the present study was 46.56 ± 2.64 years. The average age of onset of IBD was 39.03 ±2.22 years. The average duration of IBD was 6.76 ± 0.97 years.

The mean age of the patients with minimal depression was 42 ± 2.99 years, that of those with mild depression was 44.8 ± 6.65 years, that of those with moderate depression 53 ± 8.54 years, and that of those with moderately severe depression 69 ± 16 years.

Of the patients affected by minimal, moderate, and moderately severe depression, most of them were females, while mild depression affected mostly the males (Table 1). Of the patients with UC, 61.53% had minimal depression, 40% had mild depression, and 80% had moderate depression. 100% patients with UC had moderately severe depression. No patients with CD suffered moderately severe depression, with all of them having UC (Table 1).

The durations of IBD in patients with minimal depression and mild depression were 5.57 ± 1.24 years and 5.44 ± 1.46 years, respectively. The duration of IBD for patients with moderate depression was 9.4 ± 3.17 years and in patients with moderately severe depression it was 14.5 ± 0.5 years.

Patients were in clinical remission with the Harvey-Bradshaw Index (HBI) for CD < 5 and the Simple Clinical Colitis Activity Index (SCCAI) for CU < 5.

Patients with minimal anxiety had a mean age of 44 ± 3.15 years, those with mild anxiety a mean age of 55.8 ± 4.63 years, and with moderate anxiety a mean age of 45 ± 8 years.

Minimal and mild anxiety were present in balanced proportions in males and females, while moderate anxiety was present only in males (Table 1). More than half of the patients with minimal anxiety and all with moderate anxiety had UC, while a balanced proportion of patients with both CD and UC had mild depression (Table 1).

The duration of IBD in patients with minimal anxiety and mild anxiety was 5.77 ± 0.95 years, respectively 9.83 ± 2.93 years. The duration of IBD for patients with moderate anxiety was 8.5 ± 5.5 years.

Patients aged between 20–39 years reported a perception on the quality of life which was higher (i.e., 84 ± 4.1) than in the rest of the age classes (77 ± 3.4 for patients between 40–59 years and 62.5 ± 9.24 for patients over 60 years).

### 3.2. Central Tendency

The median values of calprotectin increased in direct proportion to the severity of depression, i.e., from 66.9 µg/g (0.1–291) (*p* = 0.001) for those with minimal depression to 193 µg/g (3.6–273.1) (*p* = 0.002) for those with mild depression. They were 269 µg/g (2.8–337.4) and 294.7 µg/g (143.1–446.4) for those with moderate depression and moderately severe depression (*p* = 0.03), respectively (Table 2, Figure 1a).

Calprotectin levels in mild, moderate, and moderately severe depression, 198 µg/g (2.8–446.4), were significantly higher (*p* = 0.04) compared to patients with mild depression, 66.9 µg/g (0.1–291).

The median values of calprotectin were not correlated with the severity of anxiety (*p* > 0.05). The median values were 156.1 µg/g (0.1–446.4) (*p* = 0.00001) for those with minimal anxiety and 116.2 µg/g (2.8–297.1) and 201 µg/g (143.1–259.9) (*p* = 0.1031) for those with mild and moderate anxiety, respectively. There were no cases of severe anxiety (Table 2, Figure 1b).

### 3.3. Correlations

The level of calprotectin was positively corelated with PHQ-9 (ρ = 0.416, *p* = 0.022) and negatively correlated with EQ-5D (ρ = −0.304, *p* = 0.033). Calprotectin level was not associated with GAD-7 (ρ= 0.059, *p* = 0.379).

Depression correlated with anxiety (ρ = 0.621, *p* < 0.001) and with the quality of life (ρ = −0.605, *p* < 0.001).

Calprotectin level was positively corelated with autolytic ideation (ρ = 0.557, *p* = 0.001), fatigue (ρ = 0.514, *p* = 0.002), slowness (ρ = 0.490, *p* = 0.003), and sleep disorders (ρ = 0.403, *p* = 0.014) (Figure 2a,b.)

### 3.4. Operating Characteristic Curve Analysis

In logistic regression analysis, calprotectin was an independent predictor of depression (odds ratio: 1.01,95% confidence interval: 1.002–1.3, *p* < 0.001).

An ROC analysis was performed to determine the cut-off value of calprotectin for predicting depression (Figure 3). A calprotectin level of 131 µg/g or higher predicted depression with a sensitivity of 82%, a specificity of 61%, and an accuracy of 70%, and could be part of a screening test for depression.

## 4. Discussion

This study assessed calprotectin levels according to depression severity, anxiety, and quality of life. Additionally, the study examined the correlations between the levels of calprotectin and of depression and anxiety and tried to find a cut-off value for calprotectin to screen depression.

Age significantly predicts depression and anxiety and determines a poor quality of life [34]. Similarly, in the present study, the severity of depression increased with age.

According to the meta-analyses conducted by Platt and Salk, women are predominantly affected by depression [35,36]. Liu found similar data and showed that women with IBD have predominant anxiety, depression, and sleep disturbances. He suggested that gender differences should be taken into account in the diagnosis and treatment of IBD [37]. Unlike previous studies, the current findings show no differences based on gender in depression.

Most patients with ulcerative colitis are affected by depression. However, Barberio showed that patients with Crohn’s disease have symptoms of anxiety and depression more often than patients with ulcerative colitis [38].

This study showed that the level of calprotectin is correlated with depression. Bisgaard’s review showed, with limited evidence, that there seems to be a bidirectional relationship between IBD and depression, not only in the sense of generating disorders but also in the sense of finding treatment solutions [39]. Joining the research of this axis, the present study identified a novel theme that was not previously identified, investigating calprotectin as a potential biomarker for IBD depression.

The correlation between calprotectin and depression was also highlighted by Foster, who analyzed post-mortem calprotectin in the brain. He found median calprotectin levels in the brains of patients (i.e., located in microglia) with schizophrenia and intermediate levels in those with bipolar disorder and major depression. The study concluded that elevated calprotectin levels in the brain may reflect inflammatory processes in the pathogenesis of severe mental disorders [40].

Mikocka-Walus, in a systematic review of controversies regarding the comorbidity of depression and anxiety in IBD, investigated whether they are higher in inactive versus active disease. He found that their rates are higher when the disease is active. Thus, there are conclusions that warrant a systemic approach to screening and treating, especially for active disease [41]. On the contrary, Mules found that the association between depression, anxiety, and disease activity scores was not significant but that gastrointestinal symptoms were significantly associated with symptoms of depression and anxiety [42].

The present data show the existence of a relationship between increased calprotectin levels and the severity of depression. Liśkiewicz showed a decrease in calprotectin level during his study due to better clinical results [43]. On the other hand, Melchior found elevated levels of calprotectin in a subgroup of patients (N = 34) with irritable bowel syndrome without identifying the inflammation of the colon as the cause [44].

In further analyses of the relationship between calprotectin and depression, the associations between calprotectin and depression symptoms were investigated. The highest associations were found with autolytic ideation, sleep changes, fatigue, and slowness of movement.

In the present study, it was found that calprotectin correlates with autolytic ideation. Ohlsson showed that the “leaky gut” hypothesis might help explain some of the immune activation that is frequently reported in people with ideation or autolytic attempts. The analyzed intestinal permeability markers were IFAB2 and zonulin [45]. Swart found that 20% of IBD patients have clinically significant depression (moderately severe and severe), with 5% having symptoms of severe depression (with 1% expressing suicidal ideation) [46].

The results from this study report that calprotectin correlates with fatigue. Grimstad studied fatigue in IBD and subsequently found that fatigue was associated with subjective rather than objective aspects of disease activity. More significant fatigue was associated with several symptoms of UC but not with objective markers of disease activity or the location of colon disease [47]. Another study by Grimstad showed that fatigue was found in almost 50% of patients with newly diagnosed active and untreated IBD, with this percentage being significantly higher than in healthy people. There was no significant association between fatigue and markers of disease activity, hemoglobin, or ferritin [48]. In the latest study, he found that fatigue is related to the female gender, young age, pain, and depressed mood [49].

Calprotectin correlated with slow movements. There is no similar data in the literature. Still, high levels of calprotectin have been found in patients with Parkinson’s disease, in whom the coordination of body movements is impaired by stiffness, slowness, or tremors suggesting intestinal inflammation [50]. In Lima’s study of motor disorders in Parkinson’s disease, he found that age-related dysbiosis and the resulting chronic inflammation are involved in the pathogenesis of the disease and its prevention [51]. Lorente-Picón suggested the treatment of motor symptoms through microbiota-based therapeutic strategies [52].

The present study identified a correlation between calprotectin and sleep disorders. Calprotectin level is elevated in patients with moderate and severe obstructive sleep apnea [53]). Similarly, Torun’s study showed that it could be used as a marker for the severity of obstructive sleep apnea syndrome [54,55].

Alvaro presented a systematic review of a bidirectional relationship between insomnia, anxiety, and depression, suggesting that insomnia predicts and is predicted by anxiety and depression. Insomnia might predict depression more consistently than depression predicts insomnia [56]. Disturbing sleep patterns significantly deteriorate the quality of life; three out of four depressive patients have insomnia symptoms [57]. Similarly, Spina found that woman patients with CD experience higher severity levels of depression, anxiety, and sleep-onset insomnia even when the disease is in remission. She recommends screening for depression, anxiety, and sleep disorders in CD patients ages 50 to 70 [58].

In the present study, we revealed no significant association between anxiety and calprotectin, in contrast with the findings from previous studies. Swart found that 29% of patients with IBD have anxiety (moderate and severe), of whom 14% have severe anxiety. Despite the severity of anxiety, few of these patients receive treatment or therapy for anxiety [59]. This observation agrees with the findings of Gracie, which states that the onset of the disease is associated with a 6-fold increase in the risk of anxiety. High anxiety scores are associated with the anticipation of disease activity and the escalation of therapy [60].

Derwa showed that high anxiety scores in patients with UC are associated with the decision to seek investigations. Sixty percent of investigations were found to be futile, being useful to guide clinical decision by calprotectin level [61].

The present study shows that calprotectin is correlated with the quality of life. Voiosu noted that these two simple, non-invasive tests, i.e., calprotectin and quality of life scale, are practical ways to monitor disease activity and to reduce the need for repeated endoscopic examinations [62].

Factors related to the inconsistency of data in studies on depression are related to several aspects. An important factor is related to social epidemiology and implies that the social, economic, polytic, and environmental factors in which people live, and by which they are influenced, determine a greater variability of the condition and, implicitly, of its determinants [63]. Another factor is related to the gender, i.e., women seem to be predominantly affected by depression. Among the confounding factors, it is important to mention the lifestyle and comorbidities that influence variables, but frequently are not taken into consideration. Additionally, there are factors that affect study design and work methodology in each study and different measuring instruments. The example is related to the quality of life. In this study, we used the EQ-5D, but there also exists another specific scale, the Short Inflammatory Bowel Disease Questionnaire (SIBQ).

These results must be interpreted with caution because of the limited availability of the patients who were part of the study. Even though the sample size in this study was rather limited, it can be considered as representative, at least for our region and community, where only 128 patients exist with IBD. Thus, our findings are based on one-quarter of the existing patients from South-Eastern Romania. This could be considered as a limitation of the study, alongside monocentric transversal aspects and the absence of a healthy control group or a control group with other psychiatric comorbidities. Additionally, we chose to prioritize the sensitivity over the specificity of calprotectin. Considering the search for an affordable and non-invasive biomarker for depression screening, the article raises a provocative question. The limitations of our study underscore the need for further research on large patient populations and a multicenter.

This paper represents a first step towards a more profound understanding of calprotectin’s potential as a screening biomarker for depression.

## 5. Conclusions

Testing for calprotectin level is a routine test used to diagnose and monitor patients with bowel diseases. The data presented in this study revealed that calprotectin levels are associated with the severity of depression. Moreover, a calprotectin level of 131 µg/g or higher could be part of a potential accessible screen test for depression in patients with IBD.

It is important to diagnose depression as soon as possible since its diagnosis allows for rapid intervention, contributes to reducing the symptoms, increases the therapeutic complacency, leads to better patient outcomes, and can improve the quality of life.

## Figures and Tables

**Figure 1 medicina-59-01240-f001:**
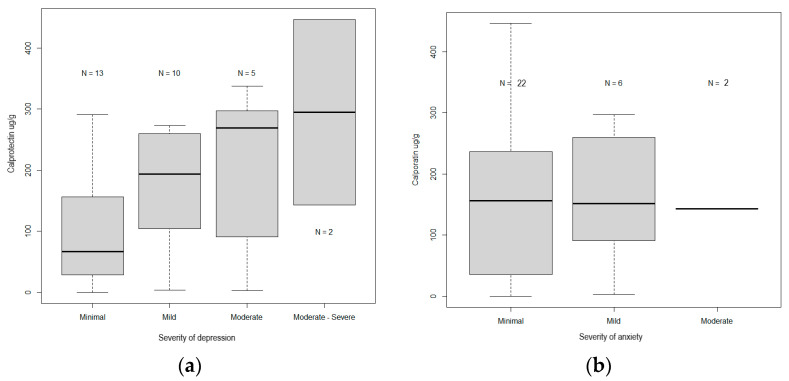
(**a**) Comparison between the levels of calprotectin and the levels of severity of depression. (**b**) Comparison of the levels of calprotectin and the levels of severity of anxiety.

**Figure 2 medicina-59-01240-f002:**
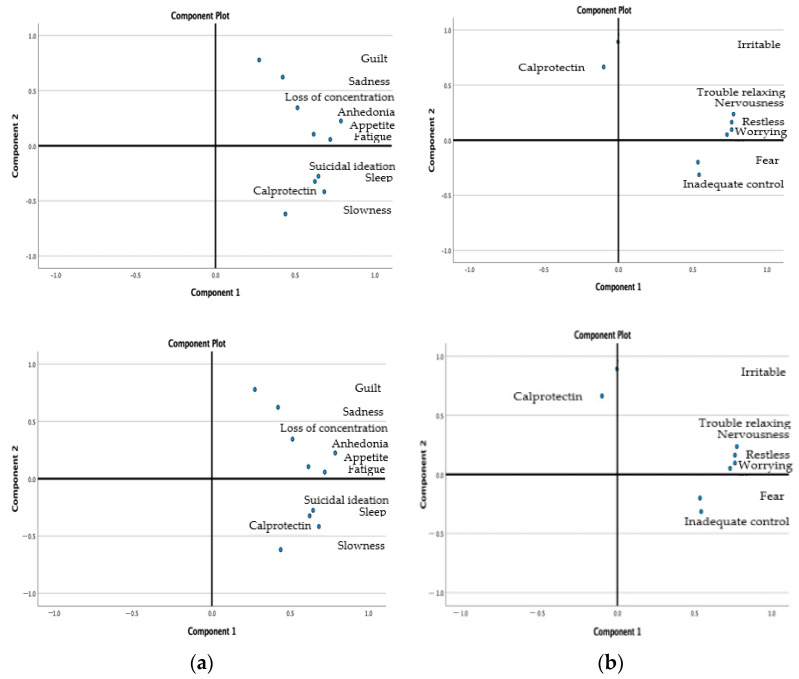
(**a**) Clusters of calprotectin with the symptoms of depression. (**b**) Clusters of calprotectin and symptoms of anxiety.

**Figure 3 medicina-59-01240-f003:**
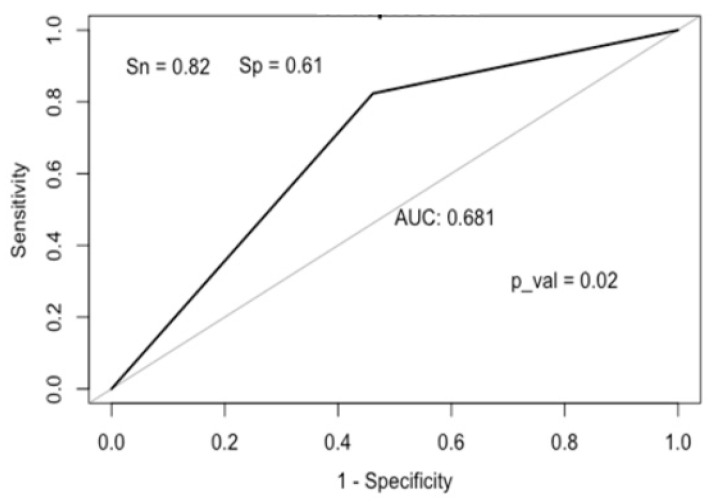
ROC curve for calprotectin.

**Table 1 medicina-59-01240-t001:** Reference characteristics and clinical data of the studied population.

	Minimal	Mild	Moderate	Moderately Severe
**Depression**				
N	13	10	5	2
Age (years)	42 ± 2.99	44.8 ± 6.65	53 ± 8.54	69 ± 16
Gender (women)	9 (69.23%)	2 (25%)	3 (60%)	1 (50%)
IBD ^1^ (UC ^2^)	8 (61.53%)	4 (40%)	4 (80%)	2 (100%)
**Anxiety**				
N	22	6	2	-
Age (years)	44 ± 3.15	55.8 ± 4.63	45 ± 8	-
Gender (women)	12 (54%)	3 (50%)	0 (0%)	-
IBD (UC)	13 (59%)	3 (50%)	2 (100%)	-

^1^ IBD-inflammatory bowel disease, ^2^ UC-ulcerative colitis.

**Table 2 medicina-59-01240-t002:** Comparison of calprotectin levels and the severity of depression and anxiety.

Calprotectin ug/g Median, Range	Minimal 1–4	Mild 5–9	Moderate 10–14	Moderately Severe 15–19
**Depression**				
N	13	10	5	2
<50 µg/g	17.3 (2.8–42.5)	11.05 (3.6–18.5)	2.80	-
50–120 µg/g	5.7 (50–66.9)	104.2	91.2	-
>120 µg/g	167 (103.3–291)	221.4 (157.4–273.5)	297.1 (269.1–337.4)	294.77 (143.1–446.4)
**Anxiety**				
N	22	6	2	-
<50 µg/g	17.9 (0.1–42.5)	3.2 (2.8–3.6)	-	-
50–120 µg/g	66.9 (50.5–106.3)	91.2	-	-
>120 µg/g	211.5 (137.6–446.4)	198.4 (104.2–297.1)	201.5 (143.1–259.9)	-

## Data Availability

Not applicable.

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
