# Peer review of "Calprotectin, Biomarker of Depression in Patients with Inflammatory Bowel Disease?"

_medicina, 2023, doi:10.3390/medicina59071240_

Round 1

Reviewer 1 Report

This work has interest in that it correlates a biochemical datum (i.e., fecal calprotectin) with levels of depressive symptoms. The work must be STRONGLY linguistically correct. The data the authors want to present is also interesting but should be presented methodologically and formally better.There are some modifications that the work needs to undergo before this paper can be further considered:

ABSTRACT

- line 27, we are missing a parenthesis before "sleep disorders."

INTRODUCTION

- Lines 38-40: In this part I should better specify the symptoms in relation to the type of IBD because so always an indistinct whole and a list of symptoms. In the CD there is no diarrhea with bleeding except for specific situations. So probably to avoid lengthening the discussion one could write that they are associated with gastrointestinal and extra-intestinal clinical manifestations;

- It should be mentioned in the introduction that given the prevalence of anxiety and depression in IBD, ad hoc psychotherapeutic techniques have been developed for these disorders. I suggest briefly mentioning discussing this recent review that addressed this important issue (https://doi.org/10.1080/27706710.2023.2181101);

- Lines 80-85: The purpose of the study is explained in an extremely general way. This study cannot presume to fully examine the relationship between calprotectin and depression, anxiety, and even quality of life. The specific outcomes of the study need to be specified;

MATERIALS AND METHODS

- It is unclear why the acronym of each questionnaire is presented in section 2.3 while it is already employed in section 2.1;

- Needless to say in the exclusion criteria patients < 18 years if you already say in the inclusion criteria ages > 18 years, it is unnecessary repetition;

- line 112: Inflammatory bowel disease should be put as acronym IBD;

- Section 2.3 why are questionnaire titles italicized for all but PHQ-9?

- Section 2.3: I think the description of the questionnaires is extremely brief: explain more about how these questionnaires are made up. I don't mean to report all the questions but at least describe the macro-areas, the domains that the questionnaires go to analyze;

- Section 2.3: Have these questionnaires already been used in other IBD studies? And if so, validated in IBD patients? If yes, cite references in that section. This is important as data to provide to the reader;

- Section 2.4: Specify how the data were displayed. is correct how you did use the median (interquartile range) because of the non-normal distribution of continuous variables but this, while a triviality, should be written in that section;

RESULTS

- Lines 140 - 144: is there a p-value connected to this data?;

- Line 143: it says "(Error! Reference ...": correct;

- 147-151: there is a missing p-value for this you say;

- Section 3 of the results: this paragraph needs to be totally rewritten grammatically and syntactically: it is really hard to read, there are several grammatical errors;

- This section is too sparse. First, it should be divided into subsections, as done for methods. One should describe the clinical and demographic characteristics of the patients. Did these patients have extra-intestinal manifestations? What was the age of the patients? What was the age at diagnosis of IBD? What were the comorbidities? Was there any difference in these parameters between CD and UC? Another section should examine the relationships between questionnaires and calprotectin. In addition, are there differences in the results related to calprotectin between CD and UC? Are there differences according to age? Are there differences between those who have or do not have extra-intestinal manifestation? Also, do you have data on biological therapy? How is this of interest? I understand that these are all subgroup analyses, purely exploratory but while specifying this in discussion they could be briefly conducted and 10-15 lines could be reserved for them in the paper;

-Is there data on disease activity of these patients? Even with a simple Partial Mayo score or HBI for Crohn's?

DISCUSSION

- Avoid phrases like "we did, we assessed." The paper should be written impersonally "This study assessed, demonstrated etc";

- Lines 176-177: but where is this in the results?

- 177-178: what data are similar? Provide numerical data of the comparison;

- The discussion needs to be rewritten from a grammatical and syntactic point of view;

- There are studies that have seen high levels of depression and anxiety even in patients with IBD totally in remission (https://doi.org/10.2174/1574887117666220328125720). This data should be cited and discussed and related to your data.

CONCLUSIONS

- Does it seem to you that a study with such large numbers could even point to calprotectin as a screen test for depression?

Good Work to the authors!

The work should be reviewed grammatically and linguistically by an English expert.

Author Response

Dear reviewer,

We highly appreciated the constructive comments received that helped us improve our manuscript. We take the opportunity to thank you for the effort!

As suggested, we reviewed the whole manuscript in order to correct the existing errors, improve the structure of the paper and offer a stronger references list of relevant studies linked to our topic. 

Please see the attachament.

Sincerely,

Melina Iordache

Reviewer 2 Report

This is a cross-sectional study in which the association between calprotectin and depression or anxiety in 30 patients with inflammatory bowel disease (IBD) was examined. Overall, this is an interesting topic highlighting the gut-brain axis significance in IBD pathophysiology with an emphasis on the potential role of calprotectin as a biomarker for detecting depression in IBD patients, but here are some of my concerns regarding this manuscript:

1.       I recommend adding the MeSH term equivalent of calprotectin in the keywords. Moreover, the authors have missed some important keywords such as depression and anxiety but instead brought some not much-related keywords such as suicidal ideation, fatigue, and sleep.  

2.       Please provide some information regarding the validity and reliability of the applied questionnaire in this study in the method section.

3.       How was the sample size of this study calculated? Please provide the effect size and power that you determined in the sample size calculation and also the formulation that has been applied for sample size estimation.

4.       Please note the inclusion and exclusion criteria in a few structured sentences.  

5.       Due to the small sample size and the cross-sectional design of this study, the results should interpret with caution and this point should be addressed in the discussion.

6.       More specifically, the manuscript lacks a thorough literature review in the introduction and discussion. In the discussion, the authors mostly focused on studies that found a positive correlation between calprotectin and anxiety or depression. I highly recommend reviewing the literature, or at least checking the following manuscripts to provide a more comprehensive discussion:

·         Melchior C, Aziz M, Aubry T, Gourcerol G, Quillard M, Zalar A, Coëffier M, Dechelotte P, Leroi AM, Ducrotté P. Does calprotectin level identify a subgroup among patients suffering from irritable bowel syndrome? Results of a prospective study. United European Gastroenterology Journal. 2017 Mar;5(2):261-9.

·         Derwa Y, Williams CJ, Sood R, Mumtaz S, Bholah MH, Selinger CP, Hamlin PJ, Ford AC, Gracie DJ. Factors affecting clinical decision-making in inflammatory bowel disease and the role of point-of-care calprotectin. Therapeutic Advances in Gastroenterology. 2018 Jan 17;11:1756283X17744739.

·         van den Brink G, Stapersma L, Vlug LE, Rizopolous D, Bodelier AG, Van Wering H, Hurkmans PC, Stuyt RJ, Hendriks DM, van der Burg JA, Utens EM. Clinical disease activity is associated with anxiety and depressive symptoms in adolescents and young adults with inflammatory bowel disease. Alimentary pharmacology & therapeutics. 2018 Aug;48(3):358-69.

·         Mules TC, Swaminathan A, Hirschfeld E, Borichevsky G, Frampton C, Day AS, Gearry RB. The impact of disease activity on psychological symptoms and quality of life in patients with inflammatory bowel disease—results from the Stress, Anxiety and Depression with Disease Activity (SADD) Study. Alimentary pharmacology & therapeutics. 2022 Jan;55(2):201-11.

Factors that may lead to inconsistent results among studies should be addressed in the discussion, providing the readers a more comprehensive point of view regarding this topic.

7.       You should cite all the tables and figures in the text.

8.       The number of cases in Figure A1 and Table A3 is not consistent. While in Figure A1, the number of patients according to the severity of anxiety is N=23, 6, and 1, respectively, it is noted N=22, 6, and 2 in Table A1, respectively.

9.       Please clarify components 1 and 2 in Figure A2.

10.    In Table A1, please provide the significance of differences among groups (p-values).

11.    In Table A1, it should be noted that the spectrum of severities is related to depression. Moreover, since you had both anxiety and depression as your measured outcomes, I highly recommend providing information similar to Table A1 for anxiety, too.

12.    Why did you transfer all tables and figures in the appendix? You can keep the main ones in the main file of the manuscript.

13.    Please include DOI for the references, if applicable.

As discussed in more detail below, due to some typos/grammatical errors, the manuscript could benefit from a revision by a native English speaker. I highly recommend rewriting some parts to be easier for readers to take the home messages.

·         Page 1, line 38: “Inflammatory bowel diseases” should be changed to “Inflammatory bowel disease”.

·         Page 1, lines 40-41: It seems something has been missed in the following sentence:

“In 2019, 4.9 million IBD cases 40 worldwide [1].”

·         Page 2, lines 72-73: 25.2% and 38.9% are attributed to the prevalence of what? Because you just mentioned depression, but noted two different percentages. Please check the following sentence:

“Depression is, also a common comorbidity in IBD, with a mean prevalence of 25.2% 72 and 38.9% in patients with active disease, respectively [21].”

·         Page 2, line 84: Please change (iii) to (iv) in the following statement:

“(iii) a cut-off value for calprotectin to screen for depression”

·         Page 2, line 89: Please remove “among” in the following sentence:

“Study participants were recruited between April and June 2021, at Constanta County Hospital, from among the patients diagnosed with IBD, i.e., CD and UC.”

·         Page 4, lines 150-151, 163 and 167: the following sentence seems to be removed:

“(Error! Reference source not found.)”

·         Page 4, Line 152: Please remove “0” after significant.

·         Page 4, line 154: The p-value has been missed.

·         Page 4, Lines 158-163 should be rewritten. 

Author Response

(The authors gave the same response as above.)

Round 2

Reviewer 1 Report

The authors made the required revisions.

Reviewer 2 Report

Dear Authors, 

Thank you for your corrections. It looks fine to me. 

Best regards